# A Simple, Fast, Sensitive LC-MS/MS Method to Quantify NAD(H) in Biological Samples: Plasma NAD(H) Measurement to Monitor Brain Pathophysiology

**DOI:** 10.3390/ijms25042325

**Published:** 2024-02-15

**Authors:** Tamaki Ishima, Natsuka Kimura, Mizuki Kobayashi, Ryozo Nagai, Hitoshi Osaka, Kenichi Aizawa

**Affiliations:** 1Division of Clinical Pharmacology, Department of Pharmacology, Jichi Medical University, Shimotsuke 329-0498, Japan; ishima.tamaki@jichi.ac.jp (T.I.); kimura_n@jichi.ac.jp (N.K.); 2Department of Pediatrics, Jichi Medical University, Shimotsuke 329-0498, Japan; mizkobayashi@jichi.ac.jp (M.K.); hosaka@jichi.ac.jp (H.O.); 3Jichi Medical University, Shimotsuke 329-0498, Japan; rnagai@jichi.ac.jp; 4Clinical Pharmacology Center, Jichi Medical University Hospital, Shimotsuke 329-0498, Japan; 5Division of Translational Research, Clinical Research Center, Jichi Medical University Hospital, Shimotsuke 329-0498, Japan

**Keywords:** LC-MS/MS, NADH, NAD^+^, validation, plasma, brain, protein precipitation, endogenous

## Abstract

Nicotinamide adenine dinucleotide (NAD) is a cofactor in redox reactions and an essential mediator of energy metabolism. The redox balance between NAD^+^ and NADH affects various diseases, cell differentiation, and aging, and in recent years there has been a growing need for measurement techniques with improved accuracy. However, NAD(H) measurements, representing both NAD^+^ and NADH, have been limited by the compound’s properties. We achieved highly sensitive simultaneous measurement of NAD^+^ and NADH under non-ion pairing, mobile phase conditions of water, or methanol containing 5 mM ammonium acetate. These were achieved using a simple pre-treatment and 7-min analysis time. Use of the stable isotope ^13^C_5_-NAD^+^ as an internal standard enabled validation close to BMV criteria and demonstrated the robustness of NAD(H) determination. Measurements using this method showed that brain NAD(H) levels correlate strongly with plasma NAD(H) levels in the same mouse, indicating that NAD(H) concentrations in brain tissue are reflected in plasma. As NAD(H) is involved in various neurodegenerative diseases and cerebral ischemia, as well as brain diseases such as mitochondrial myopathies, monitoring changes in NADH levels in plasma after drug administration will be useful for development of future diagnostics and therapeutics.

## 1. Introduction

Nicotinamide adenine dinucleotide (NAD) is an essential cofactor in redox reactions integral to cellular energy metabolism. Its oxidized (NAD^+^) and reduced (NADH) states underlie its role in facilitating cellular redox processes. Intracellular NAD(H) levels, denoting the equilibrium between NAD^+^ and NADH, represent critical metabolic indices. Their assessment is essential to understand cellular responses subsequent to pharmacological interventions and genetic modifications, illuminating consequential shifts in mitochondrial function and cellular dynamics. Recent studies have shown that sirtuins, involved in longevity, are closely related to NAD^+^ levels [1,2,3,4,5,6,7,8,9]. They are also increasingly being evaluated as markers to understand biological conditions such as obesity, diabetes, and cell differentiation [10,11].

Biochemical assays, such as enzymatic colorimetry, are affordable, highly sensitive, and widely used by researchers [3,7,9,12,13,14,15,16,17,18,19,20,21,22,23], but measurement of NAD^+^ is indirect, requires tedious enzymatic manipulation and heating, and is difficult to apply to complex samples [24]. In addition, many kits specialize in measurement of cultured cells, with few reports about homogenized tissue or blood samples. Capillary electrophoresis [25], NMR [5], HPLC-NMR [26], and UV detection HPLC [27,28,29,30,31,32,33] have been used to detect trace amounts of NAD. In recent years, MRM (Multiple Reaction Monitoring)-based approaches and other studies have also become popular [34,35,36,37,38,39,40,41,42,43,44,45,46,47,48,49]. However, a wide range of values have been reported for NAD(H) due to differences in chromatographic conditions, and many reports of NADH in plasma are below the limit of detection.

To measure nucleotides such as NAD(H), ion-pairing chromatography [29,32,35,36], hydrophilic interaction liquid chromatography (HILIC) [24,35,40,42,49], or reversed-phased chromatography [49] with alkaline mobile phases are often used to achieve nucleotide separation from biological samples. This may require a dedicated system for ion-paring, reversed-phased chromatography, or may limit solvent composition of the sample solution, according to the mobile phase in HILIC [50]. In general, conventional reversed-phased chromatography often fails to separate highly polar compounds such as NAD(H).

Furthermore, quantification of endogenous compounds in biological samples, such as NAD(H), is more complex, both analytically and in terms of validation [51,52]. For pharmaceuticals, validation needs to be performed in accordance with Bioanalytical Method Validation (BMV) guidelines issued by the Food and Drug Administration, USA (FDA) [51], but this does not apply to endogenous compounds, for which no official guidelines are currently available. The reason for this is that the matrix contains endogenous compounds that are measurement compounds, and individual differences in their content make it difficult to add known concentrations of measurement compounds to a blank matrix. It is often challenging, if not impossible, to obtain samples that do not contain any genuine biological matrix analytes or accurately known analyte concentrations, and preparation of standard samples must be handled differently, with the result that validation is not easy [52].

In this study, NAD(H) measurements in murine samples (brain, plasma, whole blood) and human cells were achieved using non-ion pairing, mobile phase conditions of water or methanol with 5 mM ammonium acetate, and simple preparation methods. Labelling with ^13^C_5_-NAD^+^, a stable isotope of NAD^+^, as an alternative internal standard, enabled validation and assessment of the analytical method according to BMV guidelines. Actual measurements on mouse samples revealed an association between brain tissue and peripheral blood.

## 2. Results

### 2.1. An LC-MS/MS Approach to Quantify NAD(H) in Biological Samples

Chromatograms of each biological sample are shown (Figure 1 and Appendix A). Chromatographic conditions allowed NAD(H) to be separated under mobile phase conditions of water or methanol containing 5 mM ammonium acetate, while minimizing the total sample run time (7 min). With our method, plasma NAD(H) was quantified at the required volume of 30 μL.

### 2.2. Method Validation

#### 2.2.1. Selectivity

Figure 1 and Appendix A show that all samples met BMV criteria, as no reactions derived from interfering substances were observed in the blank samples.

#### 2.2.2. Linearity

Correlation coefficients and regression equations for calibration curves of quality-control (QC) samples (Section 4.6) were prepared using the same matrix as biological samples and using known concentrations of analytes (Appendix A). Accuracy of each concentration of standard samples for the calibration curve obtained from the regression equation was within ±15% of the theoretical standard for NAD^+^ at more than 75% of all biological sample calibration points. Five of six (83.3%) cellular NADH calibration points met BMV criteria, and the lower limit of quantification (LLOQ) was −1.9% of the theoretical value. All coefficients of determination obtained from regression equations were above 0.9965.

#### 2.2.3. Intra-Day Accuracy

Results for the intra-day accuracy of each biological sample are shown in Appendix A, which provides the results of three repeated analyses at three concentrations: blank, low (QC low), and medium (QC med). Accuracies of NAD^+^ ranged from −8.0 to +10.7 for all samples, meeting BMV criteria (within ±15%). The accuracies of NADH met BMV criteria for brain samples with 1 µM and 10 µM spiked and cell samples with 8 µM spiked. Other samples were accurate within ±30%.

#### 2.2.4. Inter-Day Precision

For each biological sample, the inter-day precision of three repeat analyses at QC low, QC med, and high concentrations (QC high) is presented as relative standard deviation (RSD%) (Appendix A). Precision ranged from 2.0 to 13.0% in all samples, meeting BMV criteria (within 15%). In terms of NADH precision, high concentrations in whole blood and QC med and QC high in cells were <20%, ranging from 16.2 to 18.2%. All others met BMV criteria.

#### 2.2.5. Injection Carryover

After measuring the upper limit of quantification (ULOQ) of each biological sample, a blank sample was measured and compared to the LLOQ area value (Appendix A). For NADH, the response in blank sample measurement was not obtained for all biological samples. Similarly for NAD^+^, whole blood samples did not provide a blank sample response. Brain, plasma, and cell samples showed only slight reactions, with carryover of 0.01%, 0.89%, and 0.05%, respectively. Thus, all samples were well within BMV criteria.

#### 2.2.6. Sample Stability

In human cells, protein precipitation with methanol (as shown in Section 4.4.4 and Section 4.6) was performed to stop the enzymatic reaction in vivo. The solvent was then replaced with water, and samples were incubated in an autosampler (4 °C). Results were normalized by the relative area ratio at 0 h and expressed as coefficients of variation (CV: %) (Table 1).

For NAD^+^, the CV was stable within 1.13% after 48 h, and for NADH, except for 1 μM, the CV met the BMV criteria of <15%. After 48 h, the CV for the added concentration of 1 μM was 20.11%, whereas the CV after 24 h was 10.41%, meeting BMV criteria. Thus, samples in the autosampler were generally stable up to 48 h.

Next, the stability of the mouse brain and blood samples without protein precipitation treatment was investigated for storage at room temperature (RT) and 4 °C (Figure 2).

Changes after 1 h and 24 h were normalized to 0 h and evaluated. NADH concentrations in whole blood changed little at either RT: F (2,12) = 0.5188 or 4 °C, *p* = 0.6080, 4 °C: F (2,12) = 3.54, *p* = 0.0619, whereas in brain and plasma samples, NADH levels decreased rapidly over time. For NAD^+^ concentrations, all murine samples showed a rapid decline with time.

### 2.3. Absolute Quantification of NAD(H) in Biological Samples

NAD(H) was measured in mouse brain and blood samples and compared for the same individual (Figure 3). Results for brain samples were corrected for tissue weight. The results showed a strong correlation between brain tissue and plasma for NAD(H). No correlation was found between brain tissue and whole blood.

## 3. Discussion

### 3.1. An LC-MS/MS Approach to Quantify NAD(H) in Biological Samples

There are many reports of measurements for NAD^+^, but few for NADH [23,24,46]. In addition, intracellular NAD^+^ levels vary between 1 μM and 1 mM [28,38]. Because NAD^+^ and NADH are present endogenously in a wide range of concentrations [53], their simultaneous analysis is very difficult. Nucleotides tend to fragment in the source, producing interfering isobaric species. Furthermore, NAD^+^/NADH redox pairs are only one mass unit apart and show similar fragmentation profiles. Therefore, robust, reliable chromatographic separations are of paramount importance to avoid misidentification and inaccurate quantification of these important metabolites [24]. In this study, water or methanol containing 5 mM ammonium acetate was used as the mobile phase to establish simple, rapid analysis in just 7 min under non-ion pairing conditions. In addition, a simple pre-treatment process was also developed to stably extract NAD(H) in biological samples.

Methanol [24,45,54], ethanol [25], perchloric acid [24,32], and acidic acetonitrile/methanol [35] are conventionally used as a pretreatment (extraction method) for nucleotides, but with our method, ice-cold 100% methanol extraction was selected to eliminate the influence of enzymes and foreign substances in biological samples. In preliminary experiments, ionization of NADH was suppressed compared to NAD^+^, so chloroform/methanol extraction, which is commonly used in metabolome analysis to remove foreign substances, was attempted during sample preparation, but it was not effective for additional steps, so simple 100% ice-cold methanol extraction was selected. Although NADH is less abundant than NAD^+^ [8,55] and is often reported to be below the detection limit in plasma, our method was able to quantify NADH at concentrations as low as 5 nM with the required sample volume of 30 μL.

### 3.2. Method Validation

Validation must follow BMV guidelines for pharmaceuticals, but full validation is not required for endogenous compounds such as NAD(H). However, in recent years, when attention has focused on biomarker evaluation, analysis of endogenous substances has become increasingly popular, requiring highly reliable data. In this study, we attempted to follow BMV guidelines as closely as possible by using the stable isotope ^13^C_5_-NAD^+^ as an internal standard for biological matrices. LC-MS/MS, using the same stable isotope, has been performed in human blood samples by Liu et al. [43] and in mouse blood and tissue samples by Liang et al. [46]. However, while NAD^+^ was reported, but not NADH. In this study, validity of detection was verified for murine samples (brain, plasma, whole blood) and matrices derived from human cells. These results showed that both NAD^+^ and NADH fully met criteria for selectivity and carryover. Linearity, accuracy, and precision also generally met BMV criteria.

A stability evaluation of NAD(H) in vivo was conducted to determine if the process from sample collection to storage affects results quantitatively. First, in human cell samples that were reconstituted with water and incubated at 4 °C after eliminating the effect of enzymes in vivo by protein precipitation, NAD^+^ was stable beyond 48 h at all spiked concentrations, but not in samples with 1 μM NADH. Redox-active species such as ascorbic acid, uric acid, and glucose reportedly interfere with NADH measurement [56]. It is possible that there was some water-soluble material that was not removed by this simplified protein precipitation method, which interfered with NADH measurements over time.

Next, results of mouse samples without protein precipitation showed that, although the concentration of NADH in whole blood did not change with time, concentrations of NADH in brain and plasma and NAD^+^ in all mouse samples decreased rapidly with time, suggesting that NAD(H) is degraded by enzymes in vivo even at room temperature and at 4 °C. Demarest et al. reported that NAD^+^ decreases, but NADH increases, after 1 h when erythrocyte samples are left at room temperature [42]. Airhart et al. reported that NAD^+^ is unstable in room-temperature blood samples. Measured levels of NAD^+^ spiked into thawed, room-temperature whole blood degraded by as much as 50% in 10 min. They reported that, even when blood samples were placed on wet ice, NAD^+^ was reduced by 3–4% in 10 min [44]. In summary, these results suggest that accurate NAD(H) measurement requires either storage at very low temperatures immediately after sample collection or inhibition of degradative enzyme(s) with methanol or some other reagent that does not affect subsequent measurements.

### 3.3. Absolute Quantification of NAD(H) in Biological Samples

There have been reports of NADH and indirectly measured NAD^+^ in human plasma using an enzymatic assay [57], but none in mouse plasma samples [53]. Although there are reports of measuring only NAD^+^ in whole blood samples [16,17,18], there are no reports of measuring NADH. Majamaa et al. measured NAD(H) in erythrocytes using enzymatic analysis, but reported that NADH was below the detection limit [58].

We performed cross-validation with LC-MS/MS and measured mouse blood samples and brain tissue using colorimetric quantification as a preliminary experiment. First, samples were assayed after protein removal treatment with a 10-kDa-cutoff column according to the kit’s instructions, but the influence of endogenous enzymes and interference of small molecule inhibitors during manipulation were considered. Next, to remove effects of endogenous enzymes, measurements were also performed on samples that had undergone protein precipitation treatment with methanol, as in LC-MS/MS, and were reconstituted with water so as not to affect reagents used in the kit. However, in both plasma and whole blood, NADH was below the detection limit, probably due to increased dilution. Also, because of the complexity of this method, LC-MS/MS is simpler and more accurate.

Lu et al. also compared cycling assay buffers when examining extraction methods for LC-MS, but measured NAD(H) values differed depending on the extraction buffer [35]. They recommended acetonitrile/methanol with 0.1 mol/L formic acid (FA), but 80% methanol was more efficient for extraction than 0.02 mol/L FA. Different compositions of extraction buffers can change NADH to NAD^+^ (or vice versa in enzyme assay buffers), so this should be kept in mind when measuring NAD(H) (especially trace amounts of NADH). Clement et al. measured NAD(H) in human plasma using LC-MS/MS [45], but there have been no reports of studies measuring NADH in mouse plasma [53].

Comparison of measured data in the same mouse showed a strong correlation between brain tissue and plasma for both NAD(H). Compared to NADH in plasma (on the order of tens of nmol/L), NADH in whole blood is an order of magnitude higher, and considering stability during sample storage, we thought whole blood would be easier to measure, but there was no correlation with the brain. It may be possible to do this with whole blood samples by correcting for hematocrit values and other parameters, but plasma is better suited for this method given the added difficulty of these manipulations. These results suggest that NAD(H) concentrations in the brain are reflected in plasma samples. Therefore, it may be possible to estimate brain NAD(H) concentrations by measuring plasma NAD(H) concentrations.

Sharma et al. reported that NADH-reducing stress is associated with the severity of mitochondrial disease [59]. Currently, biomarkers to diagnose mitochondrial myopathy are lactate and its precursor, pyruvate, which are alternative markers because NAD(H) in mitochondria cannot be measured directly. However, some patients do not have elevated lactate/pyruvate levels. Accordingly, the ability to directly measure plasma NAD(H) is expected to facilitate diagnosis of these mitochondrial encephalopathies and other diseases in the future.

Furthermore, changes in NAD(H) levels are not limited to mitochondrial myopathy, but are also implicated in neurodegenerative diseases [12] (Alzheimer’s disease, Parkinson’s disease, Huntington’s disease, amyotrophic lateral sclerosis [60]), cerebral ischemia [30], metabolic disorders [6,61] (diabetes, obesity, liver fatty disease, non-alcoholic fatty liver disease, kidney disease, heart disease), cancer, and other diseases [10,11].

Monitoring changes in NAD(H) levels after drug administration is expected to be useful in the development of therapeutic strategies, as NAD(H) levels are also associated with gene expression, DNA repair [61,62], cell differentiation, oxidative stress [7,13], inflammation [30,63], and apoptosis [64].

Although mice were used for brain and blood samples in this study, NAD(H) exhibits variation among other mammalian tissues [53] and should be validated in other tissues and clinical samples in the future.

## 4. Materials and Methods

### 4.1. Chemicals and Reagents

β-nicotinamide adenine dinucleotide (NAD^+^), β-nicotinamide adenine dinucleotide, and reduced disodium salt hydrate (NADH) were purchased from Sigma-Aldrich (St. Louis, MO, USA). Nicotinamide Adenine Dinucleotide, NH_4_ Salt [Riboce-^13^C_5_, 98%] (^13^C_5_-NAD^+^) was purchased from Cambridge Isotope Laboratories (Tewksbury, MA, USA). Other solvents were from FUJIFILM Wako Pure Chemical (Osaka, Japan).

### 4.2. Animals

Eight-week C57BL/6j mice were purchased from Takasugi Experimental Animals Supply Co., Ltd. (Kasukabe, Japan), and blood and brain samples were collected under isoflurane anesthesia. Blood samples were collected from the inferior vena cava and coagulation was immediately inhibited with MiniCllect^®^Tube EDTA-2K (Greiner, Kremsmünster, Austria) on ice to make whole blood samples. Half of each whole blood sample was transferred to a separate tube and centrifuged at 3000× *g*, 4 °C for 5 min. Supernatants were then transferred to separate tubes and used as plasma samples. Brain samples were also frozen in liquid nitrogen immediately after collection. All samples were stored at −80 °C.

### 4.3. Human Cells

Five cell-line samples were used: three lines of healthy human samples and two lines of patient samples. Three lines of fibroblasts were purchased: two lines from PromoCell Company (#C-12300, GmbH, Heidelberg, Germany) and one from Lonza Japan (# CC-2509, Tokyo, Japan). Dermal fibroblasts derived from a patient with mitochondrial disease (two lines) were provided by the Department of Pediatrics, Jichi Medical University [65].

### 4.4. Sample Extraction

#### 4.4.1. Mouse Brain Tissue

Brain tissue samples were uniformly pulverized in a frozen state with a freeze crusher (SK Mill-200, Tokken, Chiba, Japan). Crushed tissue samples were weighed into 2 mL hard tubes that were kept frozen, and 500 μL of cold methanol containing 729 nM ^13^C_5_-NAD^+^ and 5 mm stainless steel beads were added to tissue samples, homogenized, and centrifuged (15,000 rpm, 4 °C for 15 min). Aliquots of supernatant of 27 μL of supernatant were transferred to other tubes and evaporated in a vacuum concentrator. Samples were reconstituted in 60 μL LCMS grade water, sonicated for 1 min on ice, vortexed, and centrifuged (15,000 rpm for 15 min 4 °C). An amount of 55 μL of supernatant was transferred to vials for analysis.

#### 4.4.2. Mouse Plasma

Plasma samples of 27 µL, to which 75 µL of cold methanol containing 72.9 nM ^13^C_5_-NAD^+^ had been added, were vortexed for 30 s and then centrifuged (15,000 rpm for 15 min at 4 °C). Supernatant aliquots of 65 μL were transferred into other tubes and evaporated in a vacuum concentrator. Samples were reconstituted with 35 μL of LCMS-grade water, sonicated for 1 min on ice, vortexed, and centrifuged (15,000 rpm for 10 min 4 °C). Supernatant aliquots of 30μL were transferred into vials for analysis.

#### 4.4.3. Mouse Whole Blood

Whole blood samples of 18 μL, to which 150 μL of cold methanol containing 364 nM ^13^C_5_-NAD^+^ had been added, were vortexed for 30 s and then centrifuged (15,000 rpm for 15 min at 4 °C). An amount of 60μL of supernatant was evaporated in a vacuum concentrator. Samples were reconstituted in 60 μL LCMS grade water, sonicated for 1 min on ice, vortexed, and centrifuged (15,000 rpm for 15 min 4 °C). Supernatant aliquots of 55 μL were transferred to vials for analysis.

#### 4.4.4. Human Cells

To extract NAD(H) from fibroblasts in 60 mm dishes, cells were washed twice with iced PBS, and 500 μL of extraction solvent (cold methanol containing 72.9 µM ^13^C_5_-NAD^+^) were immediately added. Cell extracts were stored at −80 °C until the next step. Immediately prior to LC-MS/MS analysis, cell extracts were sonicated for 5 min on ice and centrifuged (15,000 rpm for 15 min at 4 °C). An amount of 50 μL of supernatant was transferred into another tube and evaporated in a vacuum concentrator. Samples were reconstituted with 50 μL of LCMS-grade water, vortexed, and centrifuged (15,000 rpm, 15 min at 4 °C). Supernatant aliquots of 50 μL of were transferred into vials for analysis.

### 4.5. Instrumentation and Analytical Conditions

NAD(H) concentrations were analyzed by LC-MS/MS (LCMS-8060 NX System, Shimadzu, Kyoto, Japan). For LC analyses, a Shim-pack GIST C_18_ analytical column (50 × 2.1 mm 2 µm HSS) was used. The column oven and autosampler were set to 35 °C and 4 °C, respectively. Mobile phase A consisted of 5 mM ammonium acetate in water, and mobile phase B was 5 mM ammonium acetate in methanol. The flow rate was set to 0.4 mL/min, and the injection volume was 3 μL. The gradient program was as follows: 0 to 1 min, %B = 1.5; 1 to 3 min, %B = 1 to 95% gradient; 3 to 5 min; 5.1 to 6 min, %B = 1.5. Probe position was +2.5 mm. NAD^+^ and NADH were detected in ESI-positive mode. MS/MS conditions were as follows: nebulizer gas flow (3 L/min), heating gas flow (10 L/min), interface temperature (400 °C), desolvation temperature (650 °C), heat block temperature (400 °C), and drying gas flow (10 L/min). Collision energies were 46 V for NAD^+^, 19 V for NADH, and 43 V for ^13^C_5_-NAD^+^. NAD^+^, NADH, and ^13^C_5_-NAD^+^ were observed at m/z 664.0 > 136.1, m/z 666.0 > 649.2 and 669.0 > 136.2, respectively. NAD(H) concentrations of samples were quantified by the area ratio with the interna standard reagent (^13^C_5_-NAD^+^) added to the samples.

### 4.6. Preparation of Calibration Standards and QC Samples

Stock solutions of calibration solutions NAD(H) were prepared at 1 mM in water, and internal standard (^13^C_5_-NAD^+^) was prepared at 72.9 μM in water and then stored at −80 °C. Calibration solutions were mixed just prior to sample preparation. Pooled methanol supernatants were used as matrices for each biological sample. After each stock solution was added to the matrix, a stepwise dilution was made, and the matrix was reconstituted with water as in the sample treatment, which was used as a calibration standard. QC samples were prepared in the same manner.

### 4.7. Method Validation

#### 4.7.1. Selectivity

To evaluate selectivity, blank samples and LLOQ samples were prepared using 5–6 batches of each biological sample. In the blank samples and LLOQ peak shapes of measured compounds were confirmed to be normal and not close to interference peaks.

#### 4.7.2. Linearity

Each calibration curve was prepared using the same matrix as that in the actual samples. The calibration curve consisted of a blank, a zero sample (internal standard-added blank), and QC samples (calibration points) at 4 to 6 concentrations including LLOQ, for which regression equations and correlation coefficients were calculated. Accuracies of all QC concentrations calculated from the regression equation were within ±20% of theoretical values for LLOQ and within 15% for other than LLOQ, confirming that more than 75% of the calibration points met the criteria.

#### 4.7.3. Intra-Day Accuracy

Intra-day accuracy was assessed by analyzing three replicates of QC samples in all matrices. Mean accuracy (%) between the nominal and mean measured concentration was calculated to evaluate accuracy. Intra-assay accuracy was determined by calculating the variability using the intra-assay coefficient of variation (%CV). Bias should be within ± 15% and the CV should be ≤ 15% for all tested concentration levels, except for the LLOQ, where ± 30% and ≤ 30% are accepted.

#### 4.7.4. Inter-Day Precision

Inter-day precision was determined in biological samples by analyzing three replicates of the above-described QC samples in three separate analytical runs. The relative standard deviation (RSD%) between the measured concentration was calculated to evaluate precision. Precision should be within ± 15% and the CV should be ≤ 15% for all tested concentration levels, except for the LLOQ, where ± 20% and ≤ 20% are accepted.

#### 4.7.5. Injection Carryover

Carryover was investigated by injecting a blank sample after the upper limit of quantification (ULOQ) sample. The response at the retention time of NAD(H) was compared with the response of NAD(H) in an LLOQ sample. Carry-over should not exceed 20% of the response at the LLOQ.

#### 4.7.6. Stability of Samples before and after Protein Precipitation

Stability of NAD(H) was examined in all matrices. For human cells, it was evaluated at 6 concentration levels (0–10 µM), and for mouse matrices only at the QC med. For human cell matrices, stability of methanol supernatants reconstituted with water after protein precipitation by evaporation was measured in an autosampler (4 °C) for 0–48 h. Stability was determined as the relative area ratio with 0 h as 1 and was considered stable if CV% was within 15%. For mouse matrix, stability was evaluated by relative area ratio for 0–24 h at 4 °C or RT before protein precipitation.

### 4.8. Statistical Analysis

GraphPad Prism v7.04 (GraphPad Software, Inc., Boston, MA, USA) was used for statistical analysis. Mouse sample stability data were analyzed using a one-way analysis of variance (ANOVA), followed by Dunnett’s post hoc test for differences versus baseline (0 h). Pearson’s correlation was used to investigate the relationship between selected variables. *p* < 0.05 was considered significant. Coefficients of variation (CV%) and RSD% were calculated using a validated Microsoft Excel spreadsheet.

## 5. Conclusions

The importance of measuring NAD(H) for monitoring cellular function, pathology, cell differentiation, and other biological conditions has long been discussed. However, it has been difficult to measure NAD(H) accurately due to other endogenous compounds. In the present study, the use of a stable isotope as the internal standard has enabled simple, fast, and sensitive determination of NAD(H). The validity of measurements was verified according to BMV guidelines. Accurate NAD(H) measurements showed a strong correlation between brain tissue and plasma in the same mice. This suggests that brain NAD(H) monitoring may be possible using plasma.

## Figures and Tables

**Figure 1 ijms-25-02325-f001:**
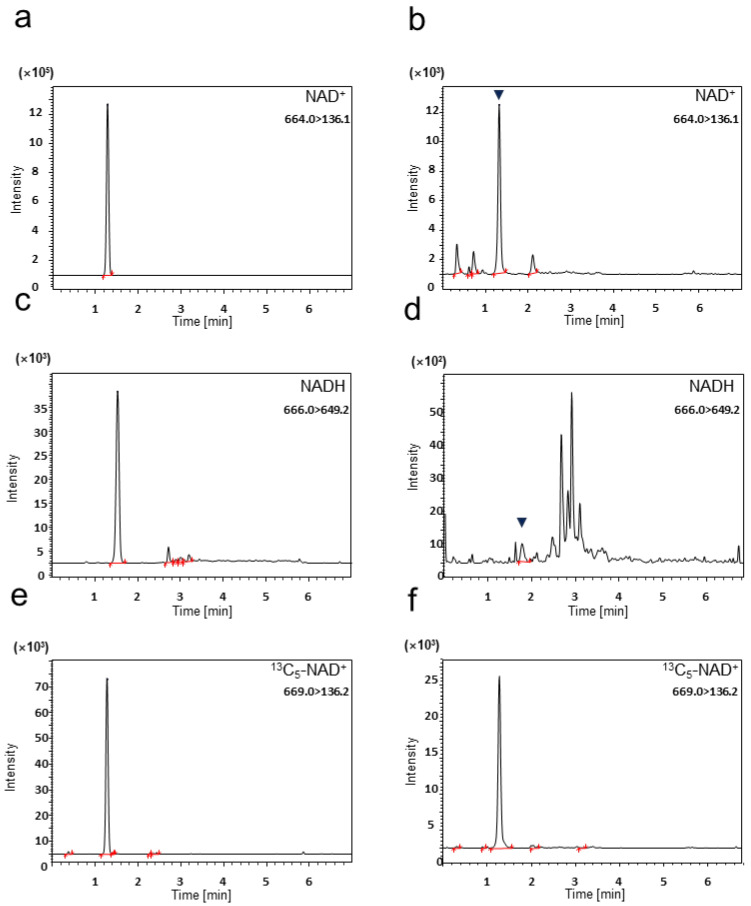
Chromatograms of NAD(H) spiked in mouse brain and plasma matrix samples. Chromatograms of (**a**) NAD^+^ in mouse brain matrix, (**b**) NAD^+^ in mouse plasma matrix, (**c**) NADH in mouse brain matrix, (**d**) NADH in mouse plasma matrix, (**e**) ^13^C_5_- NAD^+^ in mouse brain matrix, and (**f**) ^13^C_5_-NAD^+^ in mouse plasma matrix. The stable isotope, ^13^C_5_-NAD^+^, is the internal standard. For NAD^+^ and NADH, the lower limits of quantification (LLOQ) for each matrix were spiked.

**Figure 2 ijms-25-02325-f002:**
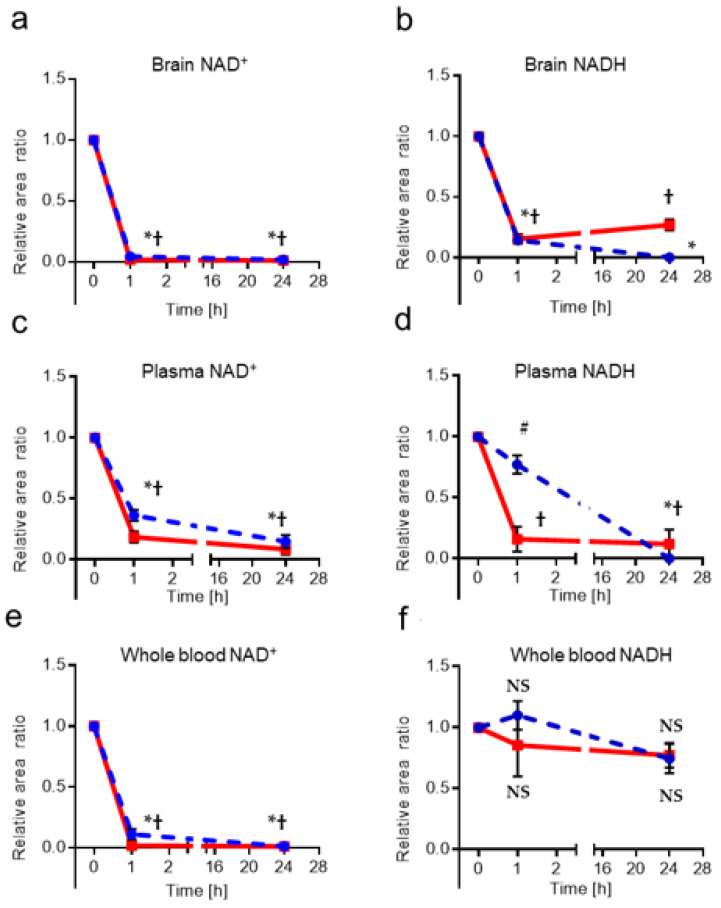
Stability at RT and 4 °C in murine samples without protein precipitation treatment. Changes after 1 h and 24 h were normalized to 0 [h] and evaluated. NAD^+^ (**a**), NADH (**b**) in brain matrix samples, NAD^+^ (**c**), NADH (**d**) in plasma matrix samples, NAD^+^ (**e**), NADH (**f**) in whole blood matrix samples (*n* = 5–6). The solid red line indicates samples at room temperature (RT) and the dotted blue line indicates samples at 4 °C. These data were analyzed with one-way analysis of variance (ANOVA), followed by Dunnett’s post hoc test for differences versus baseline (0 h). *: *p* < 0.0001 at 4 °C, #: *p* = 0.0037 at 4 °C, †: *p* < 0.0001 at RT, NS: not significant.

**Figure 3 ijms-25-02325-f003:**
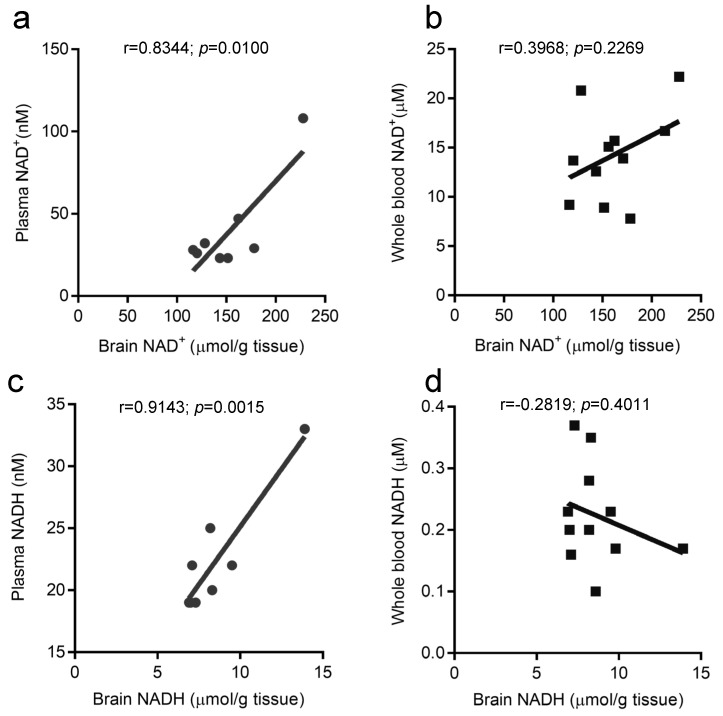
Correlation of mouse brain tissue and blood samples. Pearson correlation between brain and blood samples from the same mouse (**a**): NAD^+^, plasma (*n* = 8); (**b**): NAD^+^, whole blood (*n* = 11); (**c**): NADH, plasma (*n* = 8); and (**d**): NADH, whole blood (*n* = 11). r: correlation coefficient.

**Table 1 ijms-25-02325-t001:** Stability of protein-precipitated human cell samples in an autosampler (4 °C).

		Relative Area Ratio			
Analyte	Spiked Conc. (μM)	0 h	24 h	48 h	CV (%)
NAD^+^	0	1.00	1.01	1.01	0.51
	0.05	1.00	1.00	1.01	0.26
	0.1	1.00	1.01	1.01	0.59
	0.5	1.00	1.02	0.99	1.13
	1	1.00	0.99	0.99	0.55
	10	1.00	0.99	0.98	0.90
NADH	0	1.00	0.89	1.05	6.69
	0.05	1.00	1.03	1.22	8.88
	0.1	1.00	1.09	1.19	7.00
	0.5	1.00	1.28	1.15	9.89
	1	1.00	1.23	1.63	20.11
	10	1.00	0.76	0.72	14.97

## Data Availability

The datasets generated and/or analyzed during the present study are available from the corresponding author on request.

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
