# Peer review of "A Simple, Fast, Sensitive LC-MS/MS Method to Quantify NAD(H) in Biological Samples: Plasma NAD(H) Measurement to Monitor Brain Pathophysiology"

_ijms, 2024, doi:10.3390/ijms25042325_

Round 1

Reviewer 1 Report

Comments and Suggestions for Authors

The comments are attached in the word file 

Comments on the Quality of English Language

The English looks fine 

Reviewer 2 Report

Comments and Suggestions for Authors

The manuscript ‘A Simple, Fast, and Sensitive LC-MS/MS Method for Quantification of NAD(H) in Biological Samples: Plasma NAD(H) Measurement for Monitoring Brain Pathophysiology‘  describes a  simple, fast, and sensitive LC-MS/MS method for quantification of NAD(H) in biological samples and  have established this approach as a potential method for measuring NAD(H) in murine and plasma samples by using non-ion pairing, mobile phase conditions of water or methanol with 5 mM ammonium acetate, and simple preparation methods. The approach has emphasized straightforward sample preparation methods, streamlining the analytical process. However, LC-MS/MS methods often require expensive equipment and consumables. Hence, the cost of analysis, including instrument maintenance, may limit the method's accessibility for some research or clinical laboratories.

After going through the manuscript, I have following comment for the author.

1.      Biological samples often contain complex matrices that can affect ionization efficiency and chromatographic separation. Hence, the suggested method might be sensitive to variations in sample composition, potentially leading to variability in results. How was this point addressed in the study?

2.      Depending on the sample preparation and chromatographic conditions, the method's throughput may be limited. High sample volumes or a large number of samples may require extended analysis times. Was there any approach applied to increase the throughput oft he described method?

3.       Was the calibration curve  regularly monitored and recalibrated to  maintain accurate quantification?

Comments on the Quality of English Language

Manuscript is fine. Few minor grammatical and syntax corrections  needed. 

Round 2

Reviewer 1 Report

Comments and Suggestions for Authors

The authors have addressed the comments and suggestions in the revised manuscript, and I am happy to recommend this manuscript for publication in IJMS in the present form.